# Interplay between Autophagy and the Ubiquitin-Proteasome System and Its Role in the Pathogenesis of Age-Related Macular Degeneration

**DOI:** 10.3390/ijms20010210

**Published:** 2019-01-08

**Authors:** Janusz Blasiak, Elzbieta Pawlowska, Joanna Szczepanska, Kai Kaarniranta

**Affiliations:** 1Department of Molecular Genetics, Faculty of Biology and Environmental Protection, University of Lodz, 90-236 Lodz, Poland; janusz.blasiak@biol.uni.lodz.pl; 2Department of Orthodontics, Medical University of Lodz, 92-216 Lodz, Poland; elzbieta.pawlowska@umed.lodz.pl; 3Department of Pediatric Dentistry, Medical University of Lodz, 92-216 Lodz, Poland; joanna.szczepanska@umed.lodz.pl; 4Department of Ophthalmology, University of Eastern Finland, Kuopio 70211, Finland; 5Department of Ophthalmology, Kuopio University Hospital, 70029 Kuopio, Finland

**Keywords:** age-related macular degeneration, autophagy, mitophagy, ubiquitin-proteasome system, cellular waste elimination, proteostasis

## Abstract

Age-related macular degeneration (AMD) is a complex eye disease with many pathogenesis factors, including defective cellular waste management in retinal pigment epithelium (RPE). Main cellular waste in AMD are: all-trans retinal, drusen and lipofuscin, containing unfolded, damaged and unneeded proteins, which are degraded and recycled in RPE cells by two main machineries—the ubiquitin-proteasome system (UPS) and autophagy. Recent findings show that these systems can act together with a significant role of the EI24 (etoposide-induced protein 2.4 homolog) ubiquitin ligase in their action. On the other hand, E3 ligases are essential in both systems, but E3 is degraded by autophagy. The interplay between UPS and autophagy was targeted in several diseases, including Alzheimer disease. Therefore, cellular waste clearing in AMD should be considered in the context of such interplay rather than either of these systems singly. Aging and oxidative stress, two major AMD risk factors, reduce both UPS and autophagy. In conclusion, molecular mechanisms of UPS and autophagy can be considered as a target in AMD prevention and therapeutic perspective. Further work is needed to identify molecules and effects important for the coordination of action of these two cellular waste management systems.

## 1. Introduction

Age-related diseases are frequently associated with oxidative stress and free radicals and mitochondrial theories of aging link aging with increased production of reactive oxygen species (ROS) [1,2]. These theories have been criticized and now it is rather assumed that errors associated with every biological process increase with aging resulting in an increased amount of damaged biological molecules [3]. Proteins seem to be at the first line of the attack of the products of decreased accuracy of vital processes with age as they are the most abundant biomolecules. Mature proteins are precisely folded to be stable and functional. ROS can unfold them or contribute to their misfolding making them prone to damage and aggregation. Therefore, age-related diseases can be featured by an increased number of unfolded/damaged proteins.

Age-related macular degeneration (AMD) is a progressive and degenerative eye disease affecting the macula in the central region of the retina and leading to sight distortion. Many genetic and environmental/lifestyle factors may play a role in AMD pathogenesis, but accumulation of cellular waste and impairment in its clearing seem to be of a special significance, which was discussed by Kaarniranta et al. in 2010 [4]. This review updates and extends information and conclusions contained in that work.

AMD is a primary cause of vision loss in the elderly in developed countries. Estimated number of individuals affected by AMD in 2020 is 196 million and 288 million in 2040 [5]. Such high numbers imply high personal and public costs and an urgent need to develop efficient treatment. 

AMD can occur in dry (atrophic) or wet (exudative, neovascular) form. Dry AMD is manifested in 80–85% of all cases, but treatment options are available only to patients with wet AMD and do not address disease causes, but rely on the inhibition of vascular endothelial growth factor (VEGF) by antiangiogenic agents [6]. Aging is the most serious risk factor for AMD and the number of individuals affected increases significantly after the age of 50 years [7]. Aging is associated with intracellular accumulation of lipid and protein deposits [8]. As we will present further, such deposits are observed in AMD.

AMD is a multifactorial disease and its etiology is not completely known (Figure 1). It is suggested that oxidative stress can play a major role in the pathogenesis of AMD. The retina is susceptible to oxidative stress caused by its constant exposure to visible light and high consumption of oxygen [9]. Several life-style factors, including smoking and fat-rich diet can contribute to increased ROS production in the retina, but in general all sources of oxidative stress in the retina are unknown. Retinal pigment epithelium (RPE) is a major site of pathological alterations in AMD. In normal conditions, RPE cells regulate ion balance, secrete growth factors and maintain the blood-retina barrier. Altered bioenergetics in RPE cells manifested by reduced glycolysis and oxidative phosphorylation can contribute to AMD pathology [10]. RPE cells derived from AMD donors show an increased susceptibility to oxidative stress and produce more ROS [11]. 

Homeostasis of proteins depends on their folding, translocation and degradation (reviewed in [12]). Increased oxidative stress can cause protein misfolding and accumulation of lipid/protein aggregates observed in AMD (reviewed in [13]). Consequently, there is a need for an efficient removal of cellular waste in retinal cells to prevent AMD or slowing down its progression. Waste clearing in RPE cells includes proteasomal degradation, heterophagy, autophagy and mitophagy (reviewed in [14]). Exosomes can also be involved in waste removal in RPE cells [15]. The ubiquitin proteasome system (UPS) is mainly responsible for degradation of damaged or no longer needed proteins. Autophagy can degrade damaged organelle and may also take a part in degradation proteins when other clearance processes are failed (reviewed in [16]). RPE cells phagocytose used photoreceptors outer segments (POS) with their subsequent autophagy-lysosomal degradation [17]. The removal of POS by heterophagy occurs at the apical side of RPE cells that is linked with the photoreceptor layer [18]. Disturbances in waste clearing leads to accumulation of harmful lipid and protein aggregates that can act as a physical barrier to intracellular transport and disturb proper functioning of RPE cells.

## 2. Major Cellular Waste in Retinal Pathophysiology

### 2.1. All-Trans-Retinal

All-*trans*-retinal (atRAL) is the product of isomerization of 11-*cis*-retinal, which is an essential reaction of the visual cycle, a process occurring through a series of reactions catalyzed by membrane-bound enzymes located in photoreceptors and RPE cells [19] (Figure 2). 11-*cis*-retinal is a chromophore of rhodopsin and cone pigments. atRAL is a reactive aldehyde, whose accumulation causes toxic conjugates with proteins inducing degeneration of the mouse retina [20]. Many AMD risk factors, including aging, smoking, ultraviolet (UV) and blue light exposure, chronic inflammation and improper diet can be related to oxidative stress, but it is not known, whether oxidative stress associated with AMD belongs to the reasons or consequences of the disease or both. In any case, reduction of the stress can be important in both prevention and therapy of AMD. 

To maintain vision, atRAL released from light-activated visual pigments must be isomerized to its 11-*cis* isomer [21]. The *Rdh8* gene encodes an enzyme that reduces atRAL in rod and cone outer segments and the *Abca4* gene encodes the ATP-binding transporter of atRAL, catalyzing its movement from the inside to the outside of disc membranes of rods and cones. Mice carrying a double knock-out in the *Rdh8* and *Abca4* genes accumulated atRAL condensation products and showed altered phenotype of photoreceptors and RPE cells [20]. This phenotype is characterized by the presence of yellowish deposits called drusen, intracellular lysosomal lipofuscin, basal laminar deposits and thickening of Bruch’s membrane and is escalated by light. Intense light exposure of these mice increased atRAL levels in the retina leading to NADPH (the reduced form of nicotinamide adenine dinucleotide phosphate) oxidase-mediated overproduction of intracellular ROS [22]. Therefore, aberrant release of byproducts of the visual cycle could lead to retinal degeneration. N-retinylidene-N-retinylethanolamine (A2E) is a derivative of vitamin A, which is produced in the visual cycle [23]. A2E is also a major lipofuscin component inducing damage to RPE cells. Photosensitization of A2E leads to telomere dysfunction and DNA damage in RPE cells triggering cellular senescence, a process contributing to retinal degeneration [24,25].

### 2.2. Lipofuscin

Dysfunction in POS degradation leads to accumulation of lipid-protein aggregates resulting from oxidation of unsaturated fatty acids.

These aggregates are called lipofuscin and are composed from covalently cross-linked proteins, lipids and small amount of saccharides [26] (Figure 3). Lipofuscin accumulation reflects a weakened ability to degrade protein debris and is a hallmark of RPE cells aging [27]. One of the cytotoxic lipofuscin activity is the inhibition of degradation of oxidized proteins by binding to proteasome and lysosomal proteases. The binding of lipofuscin to the proteasome may result in the inhibition of its activity [28]. It was shown that lipofuscin-bound iron is a major intracellular source of oxidants in senescent fibroblasts so it has the ability to incorporate iron and promote the Fenton reaction [29]. We and others showed that disturbed iron metabolism might play a role in AMD pathogenesis [30,31,32]. Studies revealed that the inhibition of mitochondrial fission led to an increased formation of lipofuscin. Higher lipofuscinogenesis is also associated with downregulation of the Lon protease that is responsible for selective degradation of abnormal proteins in mitochondria [33]. The A2E fluorophore is the main hydrophobic component of lipofuscin and it is the product of the interaction between atRAL and ethanolamine. A2E was reported to accumulate in aging RPE cells and increase the expression of VEGF and some interleukins as well as other inflammatory molecules [23,34].

### 2.3. Drusen

Drusen are extracellular products located between the basal lamina of the RPE cells and collagen layer of Brunch’s membrane (Figure 4). They are composed of neutral lipids and protein derivatives, substantial cellular waste products. Large areas with small drusen are associated with the incidence of AMD [35]. Deposits accumulate when the balance between production and clearance of cellular components is disturbed. More than 40% of drusen volume is made up of lipid components dominated by esterified cholesterol and phosphatidylcholine [36]. They also contain apolipoprotein E, amyloid β, vitronectin, collagens and complement proteins [37]. The latter suggests that the formation of drusen is associated witch local inflammatory events, such as activation of the complement cascade. The impairment of the phagocytosis of the Aβ42 peptide leads to the formation of its aggregates and contributes to drusen formation. One of the components involved in Aβ42 clearing is the triggering receptor expressed on myeloid cells-2 (TREM2). It was shown that expression of TREM2 was decreased in human AMD retinas compared to control samples [38]. miRNA-34a downregulates TREM2 expression in retinas obtained from AMD donors. This downregulation is triggered by ROS and inflammatory cytokines. The level of miRNA-34a is also increased in AMD retinas leading to dysfunctional phagocytosis of Aβ42 peptides and deposits formation [38].

## 3. Waste Clearing in RPE Cells

The proteasome is the main machinery of eukaryotic cells that degrades misfolded and damaged proteins. Ubiquitin proteasome system (UPS) targets soluble proteins that are ubiquitinated prior to their degradation [39] (Figure 5).

Substrate proteins are delivered to lysosomes from the extracellular media (heterophagy) or from inside the cell (autophagy). The best described heterophagic pathway is endocytosis. Three different types of autophagy have been described in mammalian cells: macroautophagy, microautophagy, and chaperone-mediated autophagy (CMA). In macroautophagy intracellular components are sequestered by a limiting membrane to form an autophagic vacuole that then fuses with lysosomes. In microautophagy, substrates are directly internalised through invaginations of the lysosomal membrane. In contrast to this “in-bulk” degradation, in CMA, selective substrate proteins are translocated into the lysosomes one by one after binding to a lysosomal receptor LAMP-2A (lysosomal associated membrane protein 2). The ubiquitin-proteasome system (UPS) is the other major pathway for degradation of intracellular proteins inside cells. Substrates are tagged with ubiquitin that is recognised by the proteasome, the protease of this pathway. Damaged organelle and protein aggregates that cannot be degraded by proteasome are subjected by system degrading marked targets in specific organelles—autophagosomes [40]. It is called autophagy in the case of degradation of intracellular waste, heterophagy when extracellular compounds are degraded and mitophagy in the case of degradation of damaged mitochondria. Damaged proteins or organelles in both systems are detected by specific chaperone and co-chaperone proteins, and this action leads to the selection of degradation pathway [41]. Chaperone and co-chaperone also participate in the ubiquitination process. Proteins directed to the autophagy pathway undergo engulfment in autophagosomes where they are subject to degradation. Ubiquitin in both of these systems is a degradation marker, excluding ubiquitin-independent autophagy pathways [42]. Membrane proteins of damaged organelle and aggregates are marked by the attachment of ubiquitin particles. Then autophagosomal degradation system detects those ubiquitinated proteins and engulfs whole organelle and aggregates. Some of the signalling proteins, including Mfn1 and 2 (mitofusin 1 and 2), participating in mitophagy before formation of autophagosome are degraded by proteasome [43]. Moreover, UPS plays an important role in the regulation of intracellular level of ROS. Proteasome is responsible for degradation of damaged mitochondrial membrane proteins, component of photosynthetic electron transport chain. Damage in these proteins leads to abnormal increase in ROS production. Proteasomal degradation of these proteins results in restoration of normal oxidative phosphorylation (OXPHOS) process and reduction of ROS. Proteasome activation can mediate mitophagy [43]. Exosomes, extracellular vehicles released by almost all if not all eukaryotic cells, are important element of cell-to-cell communication, but they can also play a role in waste clearing [44]. Accumulated unfolded and damaged proteins are detected by unfolded protein response (UPR) and targeted for degradation by UPS or autophagy.

### 3.1. Unfolded Protein Response and Endoplasmic Reticulum-Associated Degradation

Unfolded protein response is a signaling cascade activated in response to endoplasmic reticulum (ER) stress manifested by the accumulation of unfolded and damaged proteins. UPR may also be activated by damaged mitochondria. Expression of UPR genes increases in oxidative stress and it was shown that the stress caused by cigarette smoke extracts induced UPR in RPE cells [42,43]. In ER UPR is activated by pathways initiated by three different sensors: protein kinase-like endoplasmic reticulum kinase (PERK), inositol requiring enzyme 1 (IRE1), and activating transcription factor 6 (ATF6) (Figure 6) [45,46,47,48,49]. Oxidative stress enhances the transcription of these proteins in RPE cells [43]. The UPR signalling activates transcription factors and protein kinases leading to an adaptive response involving the activation of proteasomal degradation and autophagy, chaperone induction and enhancement of antioxidant defence [50,51]. Inefficient attempts to restore homeostasis cause UPR to induce cell death by apoptosis. UPR increases apoptosis in RPE cells treated with cigarette smoke extract [52,53,54]. Therefore, UPR can play a role in AMD pathogenesis as it is involved in detecting of improper proteins and their degradation in RPE.

Endoplasmic reticulum-associated degradation (ERAD) has been considered as an integral part of UPR as many ERAD genes are controlled by UPR. However, recent studies suggest that ERAD plays a direct role in protein clearance, which is mainly underlined by IRE1α activation [55,56]. 

### 3.2. Ubiquitin Proteasome System

The process of UPS degradation is initiated by specific enzymes, E1–E3, that attach ubiquitin to the substrate in an ATP-dependent manner. Substrates with polyubiquitin tail are transported to proteasome where they undergo degradation. 

Typical 26S proteosomal complex contains three domains: a core particle (20S) with two regulatory particles (19S, caps, lids) [57]. Regulatory particles are involved in the recognition and binding of polyubiquitinated proteins and their transport to the catalytic core located at the inner surface of the 20S subunit. The core particle is composed of four heptameric annular complexes—two outer α subunits, which play a structural function and two inner β subunits with catalytic activity. The β subunits contain proteolytic active sites located on proteasomal interior surface. Energy from ATP hydrolysis is used to open lid, unfold polyubiquitinated proteins and their transport inside 20S subunit.

Oxidative stress, a major factor of AMD pathogenesis, is associated with an increased production of cellular waste, but on the other hand it may damage components of cellular waste clearing systems, which can contribute to AMD progression. A mild oxidative stress can regulate UPS activity through the stimulation of E1 and E2 to bind ubiquitin as well as the 26S proteasome [58,59]. However, strong oxidative stress can damage E1 and E2, blocking ubiquitin binding. This effect can be associated with a high concentration of oxidized glutathione, competing with E1 and E2 on their binding sites in ubiquitin [60]. Oxidative stress could also directly inactivate 26S proteasome through detachment of the 19S subunit from 20S core particle [52,61]. 

Reduction of proteasome activity during lifetime is associated with aging, another critical factor in AMD pathogenesis. Aging results in reduced expression of UPS genes and may lead to the collapse of proteasome complex, resulting in an accumulation of cellular debris [62,63,64,65,66]. These effects are correlated with weakening of the sustaining activity of the proteins quality control system and especially chaperone proteins, which play a significant role in UPS [67,68,69]. Several other effects can underline lowering of the efficacy of UPS with age, including alternations in the composition of the proteasomal subunits, reduced stability of proteasomes or their inactivation [14,65,66,67]. It has been shown that during replicative senescence the level of the β subunits decreases [70]. Proteasome activity is associated with the rate of cellular aging and the entry of cells into the senescence pathway [71]. Moreover, decreasing activity of proteasome results in increase of damaged proteins of the respiratory chain, resulting in mitochondrial dysfunction and an increase of cellular ROS level [72,73].

Insufficient activity of UPS leads to the escalation of protein deposits followed by an increase in the level of ROS and induction of chronic inflammation. Inflammation combined with constantly weakening waste cleaning system in RPE cells can induce their senescence, resulting in the development and progression of AMD [24]. 

Liu et al. showed that photooxidative stress decreased the activity of UPS in RPE. This interaction increased the expression of genes encoding proinflammatory interleukins 6 and 8 (IL-6 and -8) and downregulated the anti-inflammatory genes MCP-1 (monocyte chemoattractant protein-1) and CFH (complement factor H) [74]. Similar results obtained Qin et al. who showed that a decrease in proteasome activity in RPE led to dysregulation of the NF-κB (nuclear factor kappa-light-chain-enhancer of activated B cells) signalling pathway [75]. It was demonstrated in a mouse model that dysregulation of UPS led to retinal degeneration through photoreceptor cell death by apoptosis in a caspase-independent pathway [76]. De Carvalho. et al. suggested that proteasomal regulation may play a significant role in the control of neovascularization process important in wet AMD [77]. They showed an efficient action of UPS counteracting degenerative changes in ARPE-19 (human retinal pigment epithelial) cells through the control of the TGFβ (transforming growth factor-β) signalling. 

### 3.3. Autophagy

Autophagy degrades damaged or unneeded proteins in lysosomes. Many proteins are involved in this process, including autophagy related proteins (ATGs), mechanistic target of rapamycin (mTOR), the serine/threonine uncoordinated-51-like kinases 1 and 2 (ULK1 and ULK2), FIP-200, p62/SQSTM1, microtubule-associated protein light chain 3 (LC3) and others. Several modes of autophagy can function, including macroautophagy (usually referred as to autophagy), chaperone-mediated autophagy and microautophagy. 

Autophagy is initiated by the formation of a double-membrane vesicle, autophagosome, enclosing material to be degraded (cargo) that is delivered to the lysosome, where degradation and recycling occur [78]. 

Autophagy impairment, caused by the depletion of the core autophagy genes *ATG5* and *ATG7*, was associated with an AMD-like phenotype in mouse RPE cells. This phenotype was manifested by RPE thickening, hypertrophy or hypotrophy, pigmentary abnormalities and accumulation of oxidized proteins [79]. A2E, the main hydrophobic constituent of lipofuscin can induce damage to RPE cells through the inhibition of autophagy [80]. These findings suggest that autophagy prevents detrimental effects of A2E and inhibits the production of inflammatory factors in RPE. 

Many reports imply that oxidative stress induces autophagy in RPE cells [81,82,83,84]. Studies on RPE cells from AMD donors and mice with AMD-like phenotype suggest that autophagy increases during aging and AMD [82]. However, the autophagosomes formation in late AMD was reported to occur at lower rate than in early stages. This study also revealed that chronic oxidative stress decreased autophagic flux. Autophagy can prevent retinal cells from the damaging effects of oxidative stress [81,82]. Rapamycin induced autophagy in RPE cells and led to reduced accumulation of lipofuscin, whereas leupeptin, a blocker of autophagy, caused an increase in lipofuscin formation [82]. 

Autophagy can protect RPE cells from cell death induced by oxidative stress. RPE cells treated with paraquat, an inducer of oxidative stress and cultured with autophagy inhibitor 3-methyladenine (3-MA) showed an increase in the number of apoptotic cells compared to cells with undisturbed autophagy [81]. RPE cells under oxidative stress increased the expression of p62/SQSTM1 and autophagy [83]. Rotenone, an agent inducing mitotic catastrophe, increased autophagy and mitophagy in RPE cells protecting these cells from death [84]. 

The FIP200 protein is important in autophagy induction as it is involved in the formation of autophagosomes [85]. The conditional knockout of gene encoding FIP200 (FIP200 cKO) in mice resulted in a reduction of autophagy. These animals also displayed changes in the phenotype of RPE cells, including lipid accumulation, increasing with age. The reduction of autophagy in FIP200 cKO mice led to photoreceptors loss and retinal dysfunction [86].

Mice with knockout of ATP-binding cassette subfamily A member 4 (Abca4) and retinol dehydrogenase 8 (Rdh8) genes are characterized by impaired clearing of atRAL and they were a model of light-induced retinal degeneration [20]. Retinas of these mice were characterized by a delay in atRAL removal after light exposure [87]. Additionally, light illumination led to an increased expression of the LC3B-II and PARKIN proteins, a marker of autophagosome formation and a mitophagy regulator, respectively. These results suggest that autophagy plays an important role in protecting the retina from damage caused by light.

Injection of amyloid-β, which is a main component of drusen, to murine vitreous resulted in a upregulation of autophagy markers LC3, ATG5 and BECLIN-1. Human RPE cells treated with amyloid-β also showed autophagy induction and upregulated expression of cytokines [88].

### 3.4. Mitophagy

Damaged mitochondria are removed in a highly specific and selective pathway called mitophagy. All mechanistic aspects of this process are not exactly known and several models of it have been presented [89].

In a model proposed by Ding and Yin mitophagy is a two-steps process involving induction of canonical autophagy with ATG proteins and priming of mitochondria [90]. Canonical autophagy is underlined by several mechanisms, including AMPK activation induced by ATP depletion and suppression of mTOR mediated by mitochondrial damage resulting in ROS overproduction. These ROS induce further mitochondrial damage, which amplifies the inducing signal. Mitochondria priming could be PARKIN-dependent or independent. In the former, depolarization of mitochondrial membrane results in compromised cleavage of the PINK1 (PTEN (phosphatase and tensin homolog) induced kinase 1) protein mediated by the mitochondrial rhomboid protease PARL (presenilins-associated rhomboid-like protein, mitochondrial). Stabile PINK1 recruits PARKIN to mitochondria resulting in subsequent ubiquitination of proteins localized on the outer mitochondrial membrane. These proteins can be degraded by UPS or bound by p62/SQSTM1, which directly interacts with LC3 to bind autophagosome to faulty mitochondria. Selective mitophagy can be supported by the PI3K (phosphoinositide 3-kinase) complex activated by Ambra1. An enhanced expression of the FUNDC1 (FUN14 domain containing 1) and BNIP3L (BCL2 (B-cell lymphoma 2) interacting protein 3 like) proteins in impaired mitochondria may occur in the PARKIN-independent pathway of mitophagy. These proteins induce autophagosome to target mitochondria by a direct interaction with LC3. In this pathway, damaged mitochondria can be also targeted by Smurf1 (SMAD specific E3 ubiquitin protein ligase 1) to ubiquitinate mitochondrial proteins and induce mitophagy. ULK1 can phosphorylate ATG13 upon activation by Hsp90 (heat shock protein 90 kDa) to promote mitophagy. PINK1 is cleaved by mitochondrial proteases and degraded in the proteasome [91]. PINK1 activates PARKIN by its phosphorylation at S65 [92]. PINK1 targets the same residue in phosphorylation of ubiquitin in the S65 position [93]. Several other mechanisms of mitophagy, both PARKIN-dependent and independent could be considered.

Aging reduces the efficacy of mitophagy, which leads to the accumulation of damaged mitochondria (reviewed in [91]). Aged RPE cells have more mitochondrial DNA (mtDNA) damage and display a decreased ability to repair it as compared to young RPE [94,95,96,97]. The number of mitochondria decreased with age in rhesus RPE and aging mitochondria had an increased length and formed clusters [94]. Studies comparing RPE cells from healthy elderly with AMD patients showed that the latter had a decreased number of mitochondria [98]. RPE cells from elderly donors were more sensitive to oxidative stress than RPE from young individuals [99,100]. These findings suggest that mitophagy can be important in AMD pathogenesis. 

### 3.5. Exosomal Degradation

Exosomes are extracellular vehicles released by various cells, including epithelial cells [101]. They are an important element of the cell-to-cell communication (reviewed in [102]). They can carry out of the cells various molecules, including peptides, proteins, lipids, RNA and DNA, so they can be also considered as an important element of cellular waste clearing. In fact, waste elimination function was attributed to exosomes earlier than their communicative potential. However, some of molecules exported from cells by exosome can be only carriers of biological information. Exosome is a 9–11 protein complex having, similarly to proteasome, ring-like core structure that in humans contains nine subunits [103]. Core proteins of eukaryotic exosomes display RNase activity and belong to the RNase PH class [41]. Exosomes are present and display activity in the cytoplasm, nucleus and nucleolus. RNA degradation by exosomes is their best known and likely the main function. To perform it, the core exosomes display both exo- and endoribonuclease activities. Many aspects of exosome functioning, both as a cellular garbage bin and as an important element of the cell-to-cell communication, need further research. It is worth noting that exosomes can cooperate with autophagy in cellular waste clearing [104].

Some proteins which can be found in drusen, including annexin, enolase, CD63 are features of exosomes [15,105,106]. 

Blue light is an environmental AMD risk factor and it induces detrimental changes in the retina, which are associated with oxidative stress and overproduction of cellular waste. An increase in the proinflammatory molecules in the content of exosomes released by RPE cells after photoactive blue-light stimulation was observed [107]. Moreover, a higher level of the NLRP3 (NACHT (neuronal apoptosis inhibitor protein, class 2 transcription activator of the MHC, heterokaryon incompatibility and telomerase-associated protein 1), NLR (nucleotide-binding domain, leucine-rich repeat-containing family), and PYD (pyrin domain)-containing protein 3) inflammasome was observed in that study.

Emerging evidence suggests the role of exosomes in the activation of the complement in the immediate vicinity of RPE cells [108]. Exosomes were suggested to play a role in the occurrence and development of choroidal neovascularization, so they can be important in mechanisms of wet AMD pathogenesis and developing therapeutic strategies in this disease [109]. Exosomal proteins found in aqueous humor were postulated to be an independent molecular marker in wet AMD [110]. Exosomal miRNA, which can be important to stimulate target cells, was recently discovered in AMD [111]. Therefore, exosomes may play a multiple role in AMD pathogenesis, but uncovering the precise mechanism of this role requires further studies.

### 3.6. Heterophagy

Heterophagy, a digestion of extracellular material inside the cell, is intensively carried out in RPE cells as they constantly degrade POS to maintain the function of photoreceptors. Each RPE cell is challenged by digestion of POS from 30 to 40 photoreceptors [112]. 

Heterophagy in RPE cells involves the recognition and attachment of a POS discs, its digestion, the formation of phagosome and its fusion with lysosome and the final degradation [18]. Integrins, including ITGAV (integrin alpha V)-ITGB5 (integrin subunit beta 5) are necessary to bind POS, which ingestion requires the MERTK (c-mer proto-oncogene tyrosine kinase) protein [113]. After ingestion of extracellular cargo into vesicles they are transported to the basal end of the cell and fuse with lysosomes to degrade the cargo as it does in autophagy [114]. Heterophagy impairment can lead to accumulation of photoreceptor cell waste resulting in chronic inflammation. POS recognition by RPE may be a critical step in heterophagy as defects in this process results in death of photoreceptors [115]. 

## 4. Interplay of Autophagy and UPS in AMD

Autophagy and UPS are two main pathways to eliminate damaged and misfolded proteins from the cell. Despite the inhibition of UPS activates autophagy, these two systems were considered as independent for a long time. Some proteins, e.g. α-synuclein can be degraded at the same time by UPS and autophagy [116]. These systems share more common substrates (reviewed in [117]). However, some of them are too large to fit the proteasome. Both systems use ubiquitin as a signal molecule to label protein to degrade. This suggests that these two pathways interplay to maintain cellular proteostasis and many proteins can regulate this interplay.

Pandey et al. showed that histone deacetylase 6 (HDAC6) can be essential in the regulation of both UPS and autophagy [118]. Moreover, this regulation was shown to play an important role in the pathogenesis of various neurodegenerative diseases—HDAC6 repressed degeneration resulting from proteasome mutations in an autophagy-dependent fashion. This suggests, that HDAC1 can be important for a compensatory mechanism between UPS and autophagy.

ARPE-19 cells treated with proteasome inhibitor MG132 and chloroquine, an inhibitor of autophagy, showed an increase in ubiquitinated protein aggregates and enhanced levels of LC3-I, LC-3II and LAMP1 [119]. However, an increased level of γ-tubulin and p62/SQSTM1 was also observed, suggesting that autophagy was upregulated in that study. Chloroquine increased the levels of ubiquitinated aggregates and LC3-II and p62/SQSTM1. Prolonged inhibition of autophagy resulted in compromising of proteasome activity. These results confirm the interplay between autophagy and UPS in the retina, so waste clearing in AMD should be rather considered in the context of such interplay than in either system separately.

Ubiquitin tagging is performed with the involvement of several proteins, mainly ubiquitin ligases E1-E3 and the interaction between the RING (really interesting new gene) domain of E3 with E2 results in the final stage of attachment of ubiquitin to the protein to be degraded. It was reported that the protein EI24 (etoposide-induced protein 2.4 homolog) promoted degradation of RING E3 ligases in autophagy, which can be important in cancer transformation [120]. The gene encoding EI24 was shown to be essential for autophagy [121]. Therefore, EI24 can be the main connection between UPS and autophagy underlined by its ability to degrade RING E3 (Figure 7). It was also shown that E3 ligases, major functional proteins in UPS, can be degraded by autophagy. Although the biological relevance of EI24 has been evidenced only in cancer, it seems reasonable to consider its action as a mechanism of pathogenesis of any disorder associated with impaired clearing of cellular debris. 

p62/SQSTM1 is multifunctional protein that interacts non-covalently with ubiquitin and mediates delivery of damaged proteins for degradation in both the UPS and autophagic pathways [122,123]. Phosphorylation of p62/SQSTM1 at serine 403 leads to the attachment of ubiquitinated proteins and their targeting to degradation in autophagosomes [124]. We showed that treatment of ARPE-19 cells with the proteasome inhibitor MG-132 led to accumulation of perinuclear aggregates which rapidly colocalized with p62/SQSTM1 [125]. Based on studies in which autophagy was inhibited, it was found that p62/SQSTM1 is degraded mainly by autophagy [121]. We showed that p62/SQSTM1 was accumulated in AMD donors in macular area with a large number of drusen, supporting important role of autophagy in AMD pathogenesis [126]. p62/SQSTM1 can be also phosphorylated by ULK1 at serines 409 and 405 [127]. This phosphorylation occurs when UPS does not work properly leading to proteotoxic stress. Additionally, phosphorylation executed by ULK1 does not occur under nutrient deficiency. Proteosomal stress induced by MG-132 leads to phosphorylation of p62/SQSTM1 at serine 28 by the short form of protein kinase PINK1 (PINK1-s). This phosphorylation occurs only in cells with inhibited proteasome. Phosphorylation of p62/SQSTM1 on serine 28 is required for aggregosome formation in cells under proteasomal stress. These results suggest that PINK1-s can act as a sensor of UPS activity that can stimulate the formation of aggregosome [128].

Oxidative stress induced by H_2_O_2_ leads to inhibition of proteasome activity in RPE cells and an increase in p62/SQSTM1 expression [83]. Oxidative stress induced by cigarette smoke also inhibits the UPS pathway in RPE cells [129]. It was shown that cigarette smoke upregulated the expression of p62/SQSTM1 mRNA in ARPE-19 cells. Silencing of p62/SQSTM1 increased the accumulation of protein aggregates caused by cigarette smoke in RPE cells that showed decreased autophagy and Nrf2 (nuclear factor (erythroid-derived 2)-like 2)-mediated antioxidant response [36]. This finding suggests that p62/SQSTM1 plays a major role in the protection of RPE cells against stress induced by protein damage.

Proteins of the Hu family are RNA-binding proteins. In vertebrates there are four members of the Hu family: HuR, HuB, HuC and HuD. HuR plays a role in cellular stress response and in the regulation of cell cycle (reviewed in [130]). Treatment of ARPE-19 cell with proteasome inhibitors MG132 and AICAR (5-aminoimidazole-4-carboxamide ribonucleotide) led to activation of the HuR protein that increased p62/SQSTM1 expression [131]. 

Hsp70 is involved in proteostasis maintaining by preventing protein aggregation, refolding of denaturated and aggregated proteins and acting as a chaperone for degradation in proteasome or lysosome (reviewed in [132]). Hsp70 can also trigger the permeability of lysosomal membrane [133]. Hsp70 protects ARPE-19 cells against oxidative stress. ARPE-19 cells treated with recombinant human Hsp70 protein and then exposed to H_2_O_2_ absorbed exogenously delivered Hsp70 and localized it in late endosomes and lysosomes [134]. We observed an accumulation of ubiquitinated proteins, Hsp70, p62/SQSTM1 and Lamp-1/2 in ARPE-19 cells exposed to proteasomal inhibitors [126,135]. These protein aggregates were degraded by autophagy, when UPS was inhibited. Moreover, silencing of Hsp70 decreased the viability of RPE cells treated with proteasome inhibitors suggesting that Hsp70 is a regulator of proteostasis and may be considered in AMD therapy to eliminate protein aggregates [135]. 

Low concentration of proteasome inhibitors increased the expression of major autophagy genes *ATG5* and *ATG7* and induced LC3-I to LC3-II conversion in ARPE-19 cells [136]. UPS inhibition resulted in attenuated PI3K/Akt (protein kinase B)/mTOR signaling pathway, which inhibits autophagy [137]. 

Some kinases regulate both autophagy and UPS. An example is calcium/calmodulin-dependent protein kinase II alpha (CaMKIIα) [138]. It induces UPS activity by phosphorylation of 19S subunit of 26S proteasome. CaMKIIα directly phosphorylates Beclin-1 at serine 90 contributing to its ubiquitination and the initiation of autophagy [139]. Additionally, CaMKIIα can phosphorylate O-linked β-N-acetylglucosamine (O-GlcNAc) transferase (OGT) that promotes O-GlcNAcylation of ULK1 important for autophagosome biogenesis [140]. Protein kinase A (PKA) is an enhancer of proteasome assembly and acts as an autophagy inhibitor through phosphorylation of LC3 resulting in a reduction in recruitment of LC3 to autophagosomes [141,142]. Inhibition of proteasome activity regulates the p38 mitogen-activated protein kinase (p38/MAPK) signaling leading to autophagy inhibition [120]. p38/MAPK reduces autophagy through phosphorylation of ATG5 resulting in disturbances in fusion between LC3 and autophagosomal membrane [143]. p38/MAPK also inhibits autophagy by the phosphorylation of ULK1 that reduces its kinase activity and disrupts association with Atg13 [144]. Moreover, c-Jun N-terminal kinase (JNK) phosphorylates HuR, which is a p62/SQSTM1 activator [131]. Further studies are needed to establish the role of kinases in proteostasis, as in a perspective, they can be considered in new drugs design in therapy of diseases with impaired clearing of protein aggregates.

## 5. Conclusions and Perspectives

Drusen seem to be a particularly important substrate for waste clearing systems in AMD. However, they are extracellular objects located between RPE cell layer and Bruch’s membrane. Therefore, heterophagy and exosomes can be equally important as autophagy and UPS in waste elimination and further studies should focus on the role of these two systems in AMD. The involvement of exosomes in AMD pathogenesis is poorly known and recently several works addressed this important and new issue [15,107,109]. 

Aging and oxidative stress are major factors in AMD pathogenesis and both are associated with an increased production of cellular waste. Oxidative stress is related to enhanced concentrations of ROS that can unfold and otherwise damage proteins, which form aggregates challenging proteostasis systems. Aging is associated with a general weakening of cellular functions, including proteostasis, but the exact mechanism of this effect is not fully known (reviewed in [145]). In AMD pathogenesis, formation and accumulation of lipofuscin may be a major consequence of decreased efficacy of proteostasis with aging [29,33]. Aging in AMD pathogenesis should not be limited to cellular senescence, which is mainly studied in works on age-related changes in proteostasis. Fibroblasts obtained from centenarians displayed higher 20S proteasomal activity than cells taken from a 28-year old healthy donor, but on the other hand the lowest activity was found in the donor at 80 years [71]. Moreover, decline in autophagy with aging should be rather understood in a general way, as an increase in the amount of cellular debris with age requires increased activity of autophagy. An increase in autophagic flux was reported in aging, including retinas of patients in early stage of AMD [82]. It is suggested that an increase in autophagosome numbers in the initial stage of AMD occurs when damaged protein markers have already initiated to decrease in consequence of a decreased degradation of autophagosomes content in lysosomes [82].

Lipofuscin in the retina has been presented here as a major cellular waste occurring in AMD in the context of proteostasis, contains mainly lipids. However, protein content in lipofuscin reported in many works is high enough to constitute a substrate for cellular systems dealing with protein debris. Moreover, melanolipofuscin, another autofluorescent granule accumulating in RPE has even higher protein content [146]. Recently, Orellana-Rios et al. reported a decreased fundus autofluorescence with progression of dry AMD, concluding that lipofuscin decrease and not accumulation is linked with AMD progression [147]. This is in an apparent contrast to sever other works, but lipofuscin is not the only autofluorescent compound in the retina and results obtained on dry AMD in relatively small population (38 patients vs. 36 controls) should not be extrapolated to general properties of AMD. 

Oxidative stress results in modification of proteins, which should be subjected by proteostasis system to prevent pathological changes. However, some oxidative protein modifications are reversible and can be counteracted by cellular antioxidant systems, especially when they involve methionine and cysteine. Methionine is particularly susceptible to oxidation producing sulfide radical cation or methionine sulfoxide and the amount of these products increases in aging and some pathological conditions, like inflammation, that play a role in AMD pathogenesis (reviewed in [148]). It seems important to explore how protein oxidation is counteracted by its reverse in AMD. 

As both UPS and autophagy use ubiquitin as a major signal to indicate proteins to degrade (degron), it is important to see how the ubiquitin code changes in AMD [149]. Bortezomid (VELCAD^®^) is the first proteasome inhibitor to be used in human therapy and it is the first and only ubiquitin pathway effector to become a drug [150].

In this review, we have presented results of studies on an important role of oxidative stress and cellular waste management in AMD pathogenesis. Therefore, one can ask why there is no FDA approved drugs targeting oxidative stress and waste clearing to treat AMD? Bortezomid has been applied in cancer therapy, the only approved drugs targeting AMD are VEGF inhibitors to treat the wet form of this disease. However, many clinical trials have been undertaken to assess the role of antioxidants in AMD prevention and therapy. They are The Age-Related Eye Disease Study (AREDS), AREDS2, the Carotenoids Age-Related Eye Disease Study (CAREDS), the antioxidants, lipides essentiels, nutrition et maladies oculaires study (ALIENOR), the Taurine, Omega-3 Fatty Acids, Zinc, Antioxidant, Lutein (TOZAL) study, the Blue Mountains Eye Study, the Nutritional AMD Treatment 2 Study (NAT2), the Melbourne Collaborative Cohort Study and others (reviewed in [151]). They have resulted in recommendations and formulations, including AREDS 2 Formula recommended by the AMD experts at the National Eye Institute based on the AREDS2 study. Therefore, we are somewhere between bench and bedside with our studies on molecular mechanisms of AMD, which could be directly applied in its therapy. 

EI24 as a primary connection between UPS and autophagy may play a role in AMD pathogenesis so it is justified to address the mechanism of its action in AMD in future research. Also ubiquilins are at the crossroad between UPS and autophagy, but their role in AMD pathogenesis has not been investigated. ERAD is suggested to be a faulty protein clearance system largely independent of UPR, so its functioning in RPE cells in oxidative stress and other detrimental conditions, especially in connection with the IRE1α protein, may play a role in AMD pathogenesis and could be addressed in future research on the interplay between autophagy and UPS in AMD.

## Figures and Tables

**Figure 1 ijms-20-00210-f001:**
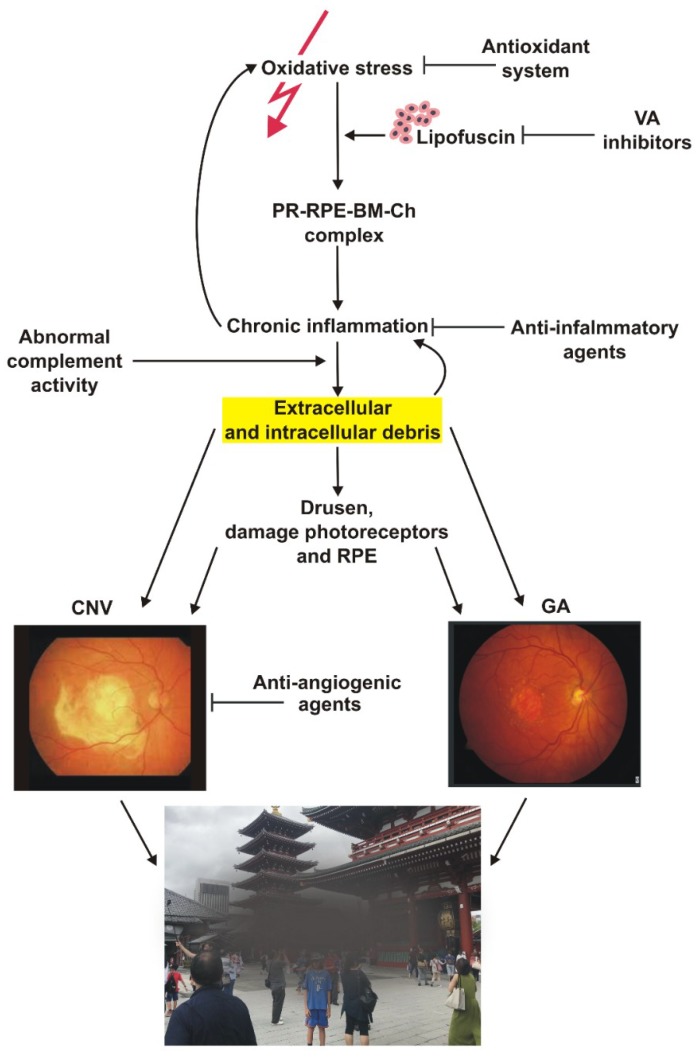
Schematic representation of the pathogenesis of age-related macular degeneration (AMD) with an important role of cellular waste (yellow highlight). Oxidative stress (red thunder) can be generated by many environmental/life style risk factors as well as yet unidentified sources. Visual cycle (VA) by-products can contribute to cellular waste. A complex interplay between oxidative stress, chronic inflammation, variants of genes encoding the complement and cellular waste clearing may lead to degeneration of retinal cells and clinically detectable AMD, which in its advanced stage may acquire the form of geographic atrophy (GA) or wet AMD, characterized by choroidal neovascularization (CNV). AMD symptoms include loss of central vision. Sharp black arrows indicate stimulation/consequences, whereas blunt black arrows—inhibition. PR—photoreceptors, RPE—retinal pigment epithelium, BM—Bruch’s membrane, Ch—choriocapillaris.

**Figure 2 ijms-20-00210-f002:**
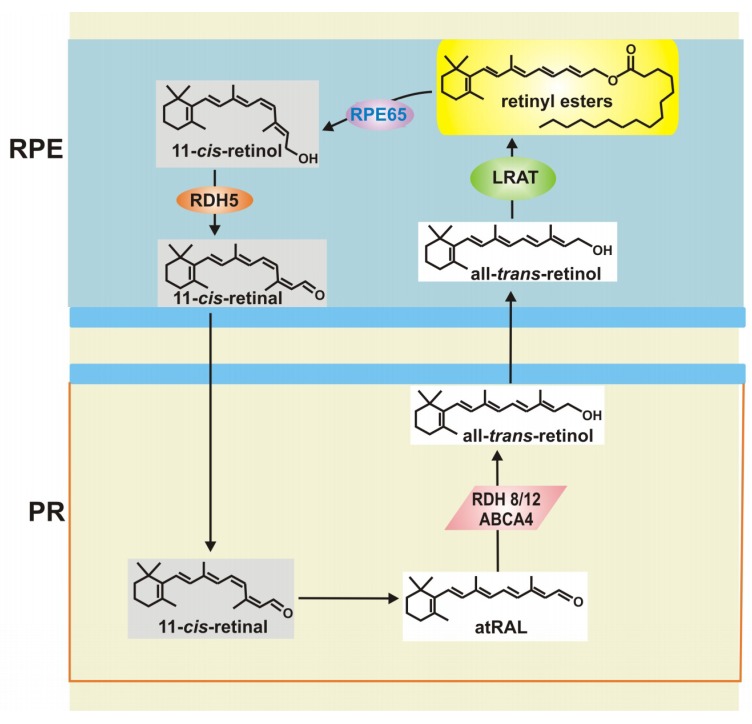
The visual cycle produces all-*trans*-retinal (atRAL), which is a major cellular waste in retinal cells. Light is absorbed by photoreceptors (PR) and causes isomerization of 11-*cis*-retinal to atRAL, which is transported and reduced to all-*trans*-retinol by ATP-binding transporter (ABCA4) and all *trans* retinal dehydrogenases RDH8/12, respectively. atRAL moves into retinal pigment epithelium (RPE), where it is converted to all-*trans*-retinyl esters by lecithin retinol acyltransferase (LRAT). RPE-specific protein (RPE65) isomerized these esters to 11-*cis*-retinol, which is then oxidized by RDH5 to 11-*cis*-retinal. Black arrows indicate a way from a compound to its derivative.

**Figure 3 ijms-20-00210-f003:**
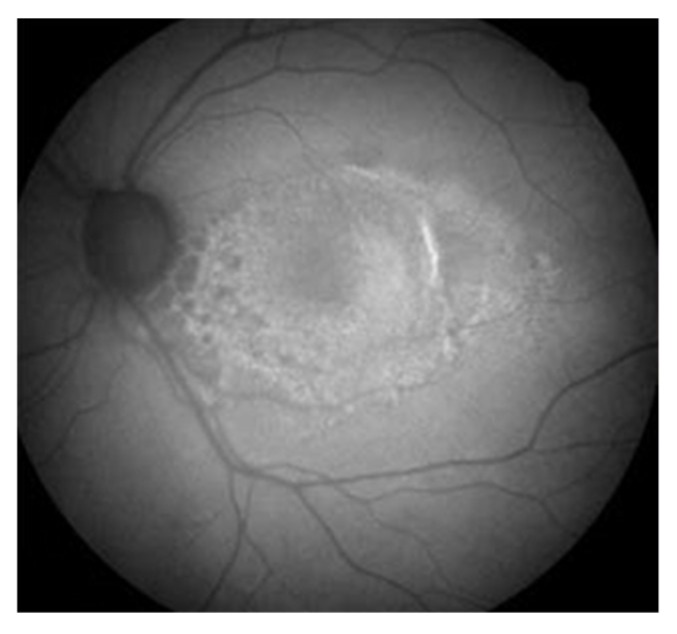
Fundus autofluorescence image from a degenerated macula indicating increased lipofuscin accumulation with increased autofluorescence signal.

**Figure 4 ijms-20-00210-f004:**
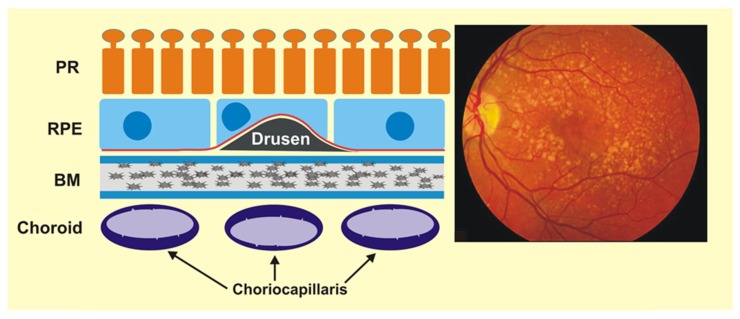
Drusen are extracellular waste located between retinal pigment epithelium (RPE) cells and Bruch’s membrane (BM), which can disturb forward vision. They are clearly visible in fundus fluorescence as scattered light stains. PR—photoreceptors.

**Figure 5 ijms-20-00210-f005:**
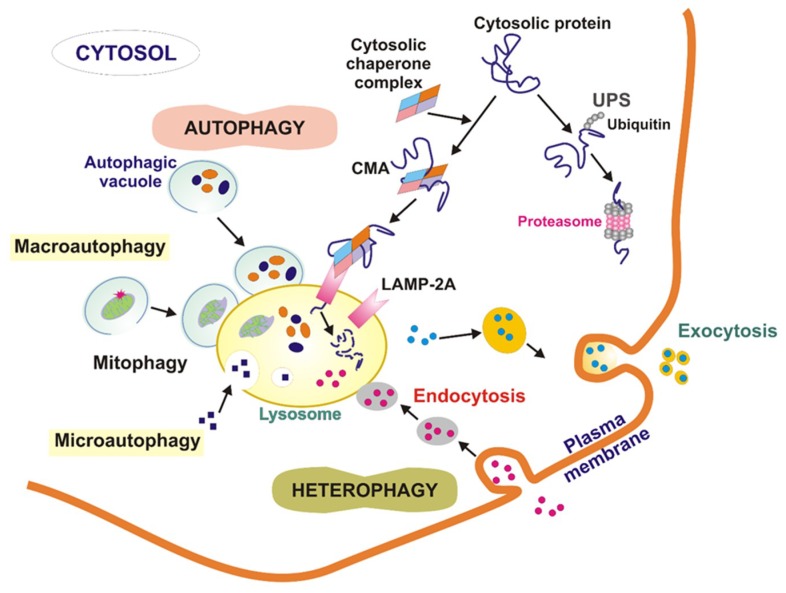
Cellular and extracellular waste clearing. Cellular waste, including misfolded, aggregated and damaged proteins as well as damaged organelles (presented as small ovals or squares of different colors) is subjected by two main machineries: ubiquitin-proteasome system (UPS) and autophagy, which can be in the form of macroautophagy, including mitophagy, microautophagy and chaperone-mediated autophagy (CMA). Unfolded proteins are a substrate for unfolded protein response (UPR, not represented here), which directs them to degradation either by autophagy or UPS. Heterophagy, which degrades extracellular debris inside the cell, is of a particular importance in retinal pigment epithelium cells and is usually carried out by endocytosis. Exosomes can transport waste material out of the cell. LAMP-2A—lysosomal associated membrane protein 2A. The black arrows indicate the sequence of events.

**Figure 6 ijms-20-00210-f006:**
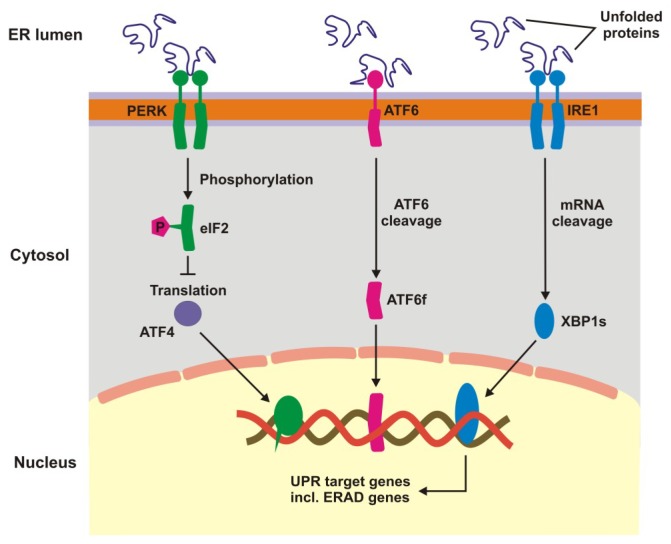
Unfolded protein response. When unfolded, misfolded and damaged proteins accumulate in endoplasmic reticulum (ER), they can induce unfolded protein response (UPR), a signaling cascade with the involvement of protein kinase-like endoplasmic reticulum kinase (PERK), inositol requiring enzyme 1 (IRE1), and activating transcription factor 6 (ATF6). This cascade leads to a stop in translation of faulty proteins, degradation of misfolded proteins and increased synthesis of chaperons involved in protein folding. If these mechanisms fall, UPR switch to pro-apoptotic response. XBP1s—X-box binding protein 1 specificity protein, eIF2—translation initiation factor 2, ERAD—ER-associated degradation, ATF6f—the transcriptional activator domain of ATF6, P—phosphate residue.

**Figure 7 ijms-20-00210-f007:**
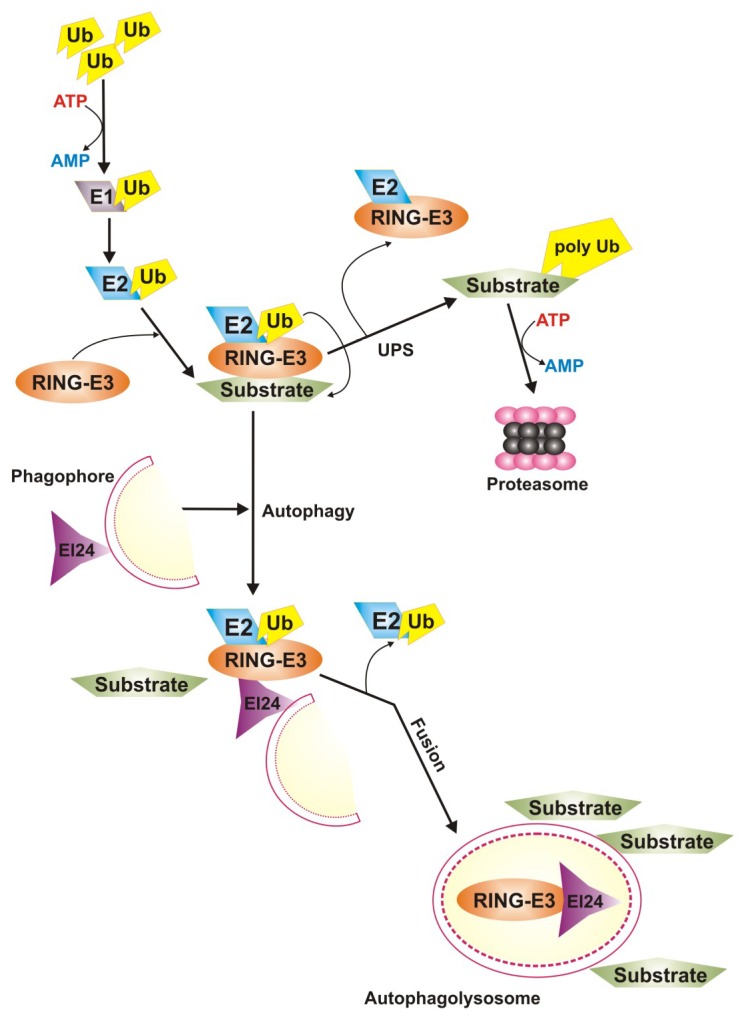
EI24 (etoposide-induced protein 2.4 homolog) is the main connection between ubiquitin-mediated proteasomal system (UPS) and autophagy. The concerted action of ubiquitin ligases E1-E3 results in ubiquitination of target proteins to label for UPS-mediated degradation. Ubiquitin chain transfer to target proteins is catalyzed by the RING-domain E3 ligases. EI24, an autophagy-inducing protein, can cause autophagy-mediated degradation of RING-domain E3 ligases. Thick arrows represent main pathways, thin arrows—side pathways.

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
