# Peer review of "Interplay between Autophagy and the Ubiquitin-Proteasome System and Its Role in the Pathogenesis of Age-Related Macular Degeneration"

_ijms, 2019, doi:10.3390/ijms20010210_

Reviewer 1 Report

This review article “Interplay between autophagy and the ubiquitin-proteasome system in age-related macular degeneration” by J. Blasiak et al. it is an interesting and extensive attempt to shed light into the role of different cellular waste sensing and management systems in the pathogenesis of AMD.   

I believe, there are several points that need to be considered before publication:

1)The statement in lines 44-45 “Therefore, age-related diseases can be featured by an increase number of unfolded/damage proteins.” Should be followed by a reference to a well-recognized research article in the AMD field (not a review), that proves that, indeed, this is the case.

2)  The chemical nature of the age-related waste accumulated in the retina is very unique.  Proteomic analysis has shown that lipofuscin granules in RPE, contains 98% lipids and less than 2% proteins, in strike contrast with lipofuscin in other tissues where it is predominantly protein-derived. Therefore, the reference 8 (line 61), ref 26 (line 135), ref 28 (line 139), ref 29 (line 140) and ref 30 (line 143) are not relevant for the retinal waste accumulated in AMD and need to be replaced or the paragraph(s) removed.

3)  There is not clear connection between the clearance via Exosomes vesicles mentioned in Figure 5, and in lines 210-212 with the exosome degradation complex described in lines 379-408.

4) The paragraph in lines 295-301, needs to be removed.

5) Line 412, should read digestion of POS from 30 to 40 photoreceptors.

6) The hypothesis that excessive oxidative stress, accumulation of misfolded proteins, or defective proteasomal and autophagic activities are not new.  Therefore, the authors should describe what interventions to solve oxidative stress and waste management are currently under clinical trials.  Specifically, if reactive oxygen species overproduction and waste accumulation it is so important for AMD, why, currently, there are no FDA approved treatments with antioxidants of autophagy inducers to treat AMD?

Author Response

1. This statement does not limit age-related diseases to AMD, so its supporting by a reference on AMD would not be appropriate. Also, we believe that this statement is a logical consequence of the first paragraph and does not need to be referenced. Moreover, we cite many references showing the presence of unfolded/damaged proteins in both AMD as well as various models of this disease further in the manuscript. Lastly, we show  some cases of waste accumulation in our original fundus photographs from AMD patients and it is accepted that that waste can contain unfolded/damaged proteins.

2. We disagree. First, waste material in AMD should not be limited to lipofuscin, despite it is likely the most useful clinical hallmark of waste accumulation in this disease due to its autofluorescence. Second, we cannot find the source information on 98% lipids/<2% proteins composition of retinal lipofuscin. Instead, we can see many reports on much higher protein content, including fundamental work of Warburton et al. (PMID: 16379024) reporting 44% of proteins in retinal lipofuscin granules. Third, melanolipofuscin that can be considered as an RPE-related variant of lipofuscin, has even higher content of proteins than basal lipofuscin (PMID: 17392682). As we know neither these and other works on this subject have been questioned and we have added a short comment on this in the Conclusions and Perspectives section:

“Lipofuscin in the retina has been presented here as a major cellular waste occurring in AMD in the context of proteostasis, it contains mainly lipids. However, protein content in lipofuscin reported in many works is high enough to constitute a substrate for cellular systems dealing with protein debris. Moreover, melanolipofuscin, another autofluorescent granule accumulating in RPE has even higher protein content [146]. Recently, Orellana-Rios et al. reported a decreased fundus autofluorescence with progression of dry AMD, concluding that lipofuscin decrease and not accumulation is linked with AMD progression [147]. This is in an apparent contrast to sever other works, but lipofuscin is not the only autofluorescent compound in the retina and results obtained on dry AMD in relatively small population (38 patients vs. 36 controls) should not be extrapolated to general properties of AMD.”

With new references:

146. Warburton, S.; Davis, W. E.; Southwick, K.; Xin, H.; Woolley, A. T.; Burton, G. F.; Thulin, C. D., Proteomic and phototoxic characterization of melanolipofuscin: correlation to disease and model for its origin. Mol. Vis. 2007, 13, 318-29.

147. Orellana-Rios, J.; Yokoyama, S.; Agee, J. M.; Challa, N.; Freund, K. B.; Yannuzzi, L. A.; Smith, R. T., Quantitative Fundus Autofluorescence in Non-Neovascular Age-Related Macular Degeneration. Ophthalmic Surg Lasers Imaging Retina 2018, 49, (10), S34-s42.

Reference 8 directly relates lipofuscin accumulation with AMD pathogenesis, although AMD is not mentioned in it. Therefore, we have replaced the sentence:

“Aging is associated with intracellular accumulation of protein and lipid deposits observed in AMD [8].”

with two sentences:

“Aging is associated with intracellular accumulation of lipid and protein deposits [8]. As we will present further, such deposits are observed in AMD.”

Reference 26 has been replaced with Sparrow 14.

Reference 28 is not related to AMD, but general properties of proteasome – we have added “In general,...” at the very beginning of the referenced sentence.

Reference 29 is closely associated with the role of lipofuscin in AMD pathogenesis, but in fact, it does not concern directly AMD and, moreover, it points at lipofuscin-bound iron. Therefore, we have changed the sequence:

“Lipofuscin is a major intracellular source of oxidants in RPE cells as has the ability to incorporate iron and promote the Fenton reaction [29].”

into

“It was shown that lipofuscin-bound iron is a major intracellular source of oxidants in senescent fibroblasts so it has the ability to incorporate iron and promote the Fenton reaction [29]. We and others showed that disturbed iron metabolism might play a role in AMD pathogenesis [30–32].”

With new references:

30. Hahn, P.; Milam, A. H.; Dunaief, J. L., Maculas affected by age-related macular degeneration contain increased chelatable iron in the retinal pigment epithelium and Bruch's membrane. Arch. Ophthalmol. 2003, 121, (8), 1099-105.

31. Gelfand, B. D.; Wright, C. B.; Kim, Y.; Yasuma, T.; Yasuma, R.; Li, S.; Fowler, B. J.; Bastos-Carvalho, A.; Kerur, N.; Uittenbogaard, A.; Han, Y. S.; Lou, D.; Kleinman, M. E.; McDonald, W. H.; Nunez, G.; Georgel, P.; Dunaief, J. L.; Ambati, J., Iron Toxicity in the Retina Requires Alu RNA and the NLRP3 Inflammasome. Cell Rep 2015, 11, (11), 1686-93.

32. Blasiak, J.; Szaflik, J.; Szaflik, J. P., Implications of altered iron homeostasis for age-related macular degeneration. Front Biosci (Landmark Ed) 2011, 16, 1551-9

Reference 30 does not support any AMD- or retina-specific expression, but it has a general meaning.

3. Figure 5 and its short immediate description present general pathways in cellular waste elimination and details of these pathways are described in subsequent sections. We do not feel to present additional figures presenting these details as they are described in detail and we do not want to overload the manuscript with textbook-like figures.

4. We are sorry that this text has been transmitted – it is from manuscript template. We have removed it.

5. Surely. We have changed that.

6. We have added the following text to the Conclusions and Perspectives section

“In this review, we have presented results of studies on an important role of oxidative stress and cellular waste management in AMD pathogenesis. Therefore, one can ask why there is no FDA approved drugs targeting oxidative stress and waste clearing to treat AMD? Bortezomid has been applied in cancer therapy, the only approved drugs targeting AMD are VEGF inhibitors to treat the wet form of this disease. However, many clinical trials have been undertaken to assess the role of antioxidants in AMD prevention and therapy. They are The Age-Related Eye Disease Study (AREDS), AREDS2, the Carotenoids Age-Related Eye Disease Study (CAREDS), the antioxidants, lipides essentiels, nutrition et maladies oculaires study (ALIENOR), the Taurine, Omega-3 Fatty Acids, Zinc, Antioxidant, Lutein (TOZAL) study, the Blue Mountains Eye Study, the Nutritional AMD Treatment 2 Study (NAT2), the Melbourne Collaborative Cohort Study and others (reviewed in [151]). They have resulted in recommendations and formulations, including AREDS 2 Formula recommended by the AMD experts at the National Eye Institute based on the AREDS2 study. Therefore, we are somewhere between bench and bedside with our studies on molecular mechanisms of AMD, which could be directly applied in its therapy.”

With new reference

151. Carneiro, A.; Andrade, J. P., Nutritional and Lifestyle Interventions for Age-Related Macular Degeneration: A Review. Oxid. Med. Cell. Longev. 2017, 2017, 6469138.

Reviewer 2 Report

The present review explains biological mechanisms that help in understanding the pathology of AMD. This work can be accepted after a few corrections.

1. More suitable title can be given that can justify the biological mechanisms that underline the AMD disease. The interplay of UPS and proteasome in AMD could be considered as one of the few mechanisms that causes AMD. Looking at the dynamic contents of the review, it will be useful to have a broad technical title.

2. The abstract should be re-written. It lacks connectivity and what to expect from this review? Moreover, a sentence like “some kinases…. and others” looks incomplete.

3. The introduction is too short and should be reorganized and re-written. Introduction can start with “Section 2: Age-related macular degeneration”. It can have flow as introducing to AMD disease- pathology-probable mechanisms of pathology.

4. Section 4.2: Ubiquitin-Proteasome System: Last few sentences “Materials and method should be described…appropriately cited” does not make any sense. Authors are requested to check.

5. Present manuscript explains various mechanisms and their role in the pathology or prevention of AMD. Authors are requested to provide a brief section on how to use the present knowledge of disease for developing new therapies.

6. The conclusion is too long and it looks like a summary of the review. Authors are requested to shorten it and provide conclusion about their understanding of the AMD pathology and how clinical outcomes can be improved using present knowledge of the disease.

Author Response

1. We have changed the title from

“Interplay between autophagy and the ubiquitin-proteasome system in age-related macular degeneration”

into

“Interplay between autophagy and the ubiquitin-proteasome system and its role in the pathogenesis of age-related macular degeneration”

2. We have changed Abstract from

Age-related macular degeneration (AMD) is a complex eye disease with many pathogenesis factor involved, including defective cellular waste management in retinal pigment epithelium (RPE). Main cellular waste in AMD are: all-trans retinal, drusen and lipofuscin, containing unfolded, damaged and unneeded proteins, which are degraded and recycled in RPE cells by two main waste clearing machineries – the ubiquitin-proteasome system (UPS) and autophagy. Recent findings show that these systems can act together with a significant role of the EI24 ubiquitin ligase in such concerted action. On the other hand, E3 ligases are essential for tagging of damaged proteins for degradation in both systems, but E3 is degraded by autophagy. Histone deacetylase 6 has been proposed to coordinate the action of both systems. The interplay between UPS and autophagy was targeted in several diseases, including Alzheimer disease. Therefore, cellular waste clearing in AMD should be rather considered in the context of such interplay rather than either of these systems singly. Aging and oxidative stress, two major AMD risk factors, reduce both UPS and autophagy. Some kinases, including calcium/calmodulin-dependent protein kinase II alpha, protein kinase A, c-Jun N-terminal kinase and others. Recently, the EI24 (etoposide-induced protein 2.4 homolog) protein has been proposed as the main connection between autophagy and UPS. Ubiquitination is critical for both UPS and autophagy, so its mechanism should be investigated in AMD, especially that it is targeted to affect UPS by an approved drug in other diseases.

into

Age-related macular degeneration (AMD) is a complex eye disease with many pathogenesis factors, including defective cellular waste management in retinal pigment epithelium (RPE). Main cellular waste in AMD are: all-trans retinal, drusen and lipofuscin, containing unfolded, damaged and unneeded proteins, which are degraded and recycled in RPE cells by two main machineries – the ubiquitin-proteasome system (UPS) and autophagy. Recent findings show that these systems can act together with a significant role of the EI24 ubiquitin ligase in their action. On the other hand, E3 ligases are essential in both systems, but E3 is degraded by autophagy. The interplay between UPS and autophagy was targeted in several diseases, including Alzheimer disease. Therefore, cellular waste clearing in AMD should be considered in the context of such interplay rather than either of these systems singly. Aging and oxidative stress, two major AMD risk factors, reduce both UPS and autophagy. In conclusion, molecular mechanisms of UPS and autophagy can be considered as a target in AMD prevention and therapeutic perspective. Further work is needed to identify molecules and effects important for the coordination of action of these two cellular waste management systems.

3. We have integrated sections 1 and 2.

4. We are sorry that this text has been transmitted – it is from manuscript template. We have removed it.

5. We have added the following fragment to the Conclusions and Perspectives section

“In this review, we have presented results of studies on an important role of oxidative stress and cellular waste management in AMD pathogenesis. Therefore, one can ask why there is no FDA approved drugs targeting oxidative stress and waste clearing to treat AMD? Bortezomid has been applied in cancer therapy, the only approved drugs targeting AMD are VEGF inhibitors to treat the wet form of this disease. However, many clinical trials have been undertaken to assess the role of antioxidants in AMD prevention and therapy. They are The Age-Related Eye Disease Study (AREDS), AREDS2, the Carotenoids Age-Related Eye Disease Study (CAREDS), the antioxidants, lipides essentiels, nutrition et maladies oculaires study (ALIENOR), the Taurine, Omega-3 Fatty Acids, Zinc, Antioxidant, Lutein (TOZAL) study, the Blue Mountains Eye Study, the Nutritional AMD Treatment 2 Study (NAT2), the Melbourne Collaborative Cohort Study and others (reviewed in 151). They have resulted in recommendations and formulations, including AREDS 2 Formula recommended by the AMD experts at the National Eye Institute based on the AREDS2 study. Therefore, we are somewhere between bench and bedside with our studies on molecular mechanisms of AMD, which could be directly applied in its therapy.”

with new reference

151. Carneiro, A.; Andrade, J. P., Nutritional and Lifestyle Interventions for Age-Related Macular Degeneration: A Review. Oxid. Med. Cell. Longev. 2017, 2017, 6469138.

6. We have removed the first paragraph from the “Conclusions and perspectives” section, but we have added the above paragraph which says something on clinical perspective. Mechanisms of pathogenesis are presented in Introduction. 

Reviewer 3 Report

Comments are printed below:

1. Abstract needs to be revised. A concise abstract discussing role of proteins and their relation to AMD is required. Also, a concrete conclusion (couple of sentences) in abstract is needed.

2. Figures 1 and 3 contains fundoscopic images. Please provides or include permissions and related references in the legend.

    Figure 2 include related references in the legend.

    Same comment/s (2 and 3) applies to other figures.

3. Authors are recommended to provide a summary table with new findings in the literature.

Author Response

1. We have changed Abstract from

Age-related macular degeneration (AMD) is a complex eye disease with many pathogenesis factor involved, including defective cellular waste management in retinal pigment epithelium (RPE). Main cellular waste in AMD are: all-trans retinal, drusen and lipofuscin, containing unfolded, damaged and unneeded proteins, which are degraded and recycled in RPE cells by two main waste clearing machineries – the ubiquitin-proteasome system (UPS) and autophagy. Recent findings show that these systems can act together with a significant role of the EI24 ubiquitin ligase in such concerted action. On the other hand, E3 ligases are essential for tagging of damaged proteins for degradation in both systems, but E3 is degraded by autophagy. Histone deacetylase 6 has been proposed to coordinate the action of both systems. The interplay between UPS and autophagy was targeted in several diseases, including Alzheimer disease. Therefore, cellular waste clearing in AMD should be rather considered in the context of such interplay rather than either of these systems singly. Aging and oxidative stress, two major AMD risk factors, reduce both UPS and autophagy. Some kinases, including calcium/calmodulin-dependent protein kinase II alpha, protein kinase A, c-Jun N-terminal kinase and others. Recently, the EI24 (etoposide-induced protein 2.4 homolog) protein has been proposed as the main connection between autophagy and UPS. Ubiquitination is critical for both UPS and autophagy, so its mechanism should be investigated in AMD, especially that it is targeted to affect UPS by an approved drug in other diseases.

into

Age-related macular degeneration (AMD) is a complex eye disease with many pathogenesis factors, including defective cellular waste management in retinal pigment epithelium (RPE). Main cellular waste in AMD are: all-trans retinal, drusen and lipofuscin, containing unfolded, damaged and unneeded proteins, which are degraded and recycled in RPE cells by two main machineries – the ubiquitin-proteasome system (UPS) and autophagy. Recent findings show that these systems can act together with a significant role of the EI24 ubiquitin ligase in their action. On the other hand, E3 ligases are essential in both systems, but E3 is degraded by autophagy. The interplay between UPS and autophagy was targeted in several diseases, including Alzheimer disease. Therefore, cellular waste clearing in AMD should be considered in the context of such interplay rather than either of these systems singly. Aging and oxidative stress, two major AMD risk factors, reduce both UPS and autophagy. In conclusion, molecular mechanisms of UPS and autophagy can be considered as a target in AMD prevention and therapeutic perspective. Further work is needed to identify molecules and effects important for the coordination of action of these two cellular waste management systems.

2. All fundus photographs have been taken by Mr. Ollie Horto from patients of Dr. Kai Kaarniranta at University of Kuopio Hospital. All other graphics/arts were performed by Ms. Monika Kicinska from Dr. Janusz Blasiak laboratory in University of Lodz.

3. We are sorry, but we cannot see how such table would look like. Few last discoveries/hypotheses in the field of interplay between autophagy and UPS have been described in the text.

Round  2

Reviewer 3 Report

No comments.